# High-throughput identification of prefusion-stabilizing mutations in SARS-CoV-2 spike

Timothy J. C. Tan [1], Zongjun Mou[2], Ruipeng Lei [3], Wenhao O. Ouyang[3], Meng Yuan [4], Ge Song [5,6,7], Raiees Andrabi [5,6,7], Ian A. Wilson [4,6,7,8], Collin Kieffer [9], Xinghong Dai[2], Kenneth A. Matreyek[10] & Nicholas C. Wu [1,3,11,12] ✉

Designing prefusion-stabilized SARS-CoV-2 spike is critical for the effectiveness of COVID-19 vaccines. All COVID-19 vaccines in the US encode spike with K986P/V987P mutations to stabilize its prefusion conformation. However, contemporary methods on engineering prefusion-stabilized spike immunogens involve tedious experimental work and heavily rely on structural information. Here, we establish a systematic and unbiased method of identifying mutations that concomitantly improve expression and stabilize the prefusion conformation of the SARS-CoV-2 spike. Our method integrates a fluorescence-based fusion assay, mammalian cell display technology, and deep mutational scanning. As a proof-of-concept, we apply this method to a region in the S2 domain that includes the first heptad repeat and central helix. Our results reveal that besides K986P and V987P, several mutations simultaneously improve expression and significantly lower the fusogenicity of the spike. As prefusion stabilization is a common challenge for viral immunogen design, this work will help accelerate vaccine development against different viruses.

SARS-CoV-2 spike (S) glycoprotein, a homotrimeric class I fusion protein, naturally exists in a metastable, prefusion conformation on the virion surface[1]. Once the receptor-binding domain (RBD) of S transitions to an 'up' state and binds to the human angiotensin-converting enzyme II (hACE2) receptor[2–4], a cascade of conformational changes is triggered to promote virus-host membrane fusion, and hence virus entry[1,5–8]. This conformational change, which involves structural rearrangement of the first heptad repeat (HR1) and central helix (CH), as well as the shedding of the S1 subunit, converts S into the postfusion conformation[5–10]. To inhibit virus entry and fusion, neutralizing antibodies target a variety of mainly conformational epitopes on the prefusion conformation of S[11–15]. Many of these conformational epitopes disappear or rearrange in the postfusion conformation, which instead can expose non-neutralizing epitopes that are immunodominant[1]. Consistently, antibody titer to the prefusion conformation has a strong correlation with neutralization potency, whereas that to the postfusion conformation does not[16]. Therefore, effective COVID-19 vaccines require S to be locked in the prefusion conformation to preserve the neutralizing epitopes.

[1]Center for Biophysics and Quantitative Biology, University of Illinois Urbana-Champaign, Urbana, IL 61801, USA. [2]Department of Physiology and Biophysics, Case Western Reserve University School of Medicine, Cleveland, OH 44106, USA. [3]Department of Biochemistry, University of Illinois Urbana-Champaign, Urbana, IL 61801, USA. [4]Department of Integrative Structural and Computational Biology, The Scripps Research Institute, La Jolla, CA 92037, USA. [5]Department of Immunology and Microbiology, The Scripps Research Institute, La Jolla, CA 92037, USA. [6]IAVI Neutralizing Antibody Center, The Scripps Research Institute, La Jolla, CA 92037, USA. [7]Consortium for HIV/AIDS Vaccine Development (CHAVD), The Scripps Research Institute, La Jolla, CA 92037, USA. [8]The Skaggs Institute for Chemical Biology, The Scripps Research Institute, La Jolla, CA 92037, USA. [9]Department of Microbiology, University of Illinois Urbana-Champaign, Urbana, IL 61801, USA. [10]Department of Pathology, Case Western Reserve University School of Medicine, Cleveland, OH 44106, USA. [11]Carl R. Woese Institute for Genomic Biology, University of Illinois Urbana-Champaign, Urbana, IL 61801, USA. [12]Carle Illinois College of Medicine, University of Illinois Urbana-Champaign, Urbana, IL 61801, USA. ✉e-mail: nicwu@illinois.edu

The rapid development of prefusion-stabilized SARS-CoV-2 S during the early phase of COVID-19 pandemic has tremendously benefited from prior studies on prefusion-stabilizing mutations in the S proteins of related betacoronaviruses, namely MERS-CoV[17,18] and SARS-CoV[18]. These studies employed a structure-based approach to identify two prefusion-stabilizing mutations (K986P/V987P, SARS-CoV-2 numbering) at the HR1-CH junction[17–19]. Due to the structural similarities among the S proteins of MERS-CoV, SARS-CoV, and SARS-CoV-2, K986P/V987P were directly applied to engineer the prefusion-stabilized SARS-CoV-2 S during COVID-19 vaccine development. For example, K986P/V987P are included in many nucleic acid- and protein subunit-based COVID-19 vaccines, such as those from Moderna[20], Pfizer-BioNTech[21], Johnson & Johnson-Janssen[22], and Novavax[23]. Subsequent studies, which also used a structure-based approach, identified additional mutations that further improve the expression and prefusion stability of SARS-CoV-2 S[24–27]. Nevertheless, identifying prefusion-stabilizing mutations using structure-based approach is time-consuming and likely not comprehensive, because it relies on low-throughput characterization of individual candidate mutants. Thus, viral vaccine immunogen design remains a challenge due to its non-trivial nature[28].

To this end, we develop here a method to identify prefusion-stabilizing mutations of SARS-CoV-2 S in a high-throughput and systematic manner, by coupling a fluorescence-based fusion assay, mammalian cell display technology, and deep mutational scanning (DMS). As a proof-of-concept, we screen all possible amino-acid mutations across the entire region spanning HR1 and CH. In addition to the K986P and V987P mutations that are used in current COVID-19 vaccines, we identify several mutations that simultaneously improve expression and stabilize the prefusion conformation of both membrane-bound and soluble S. In this regard, our method circumvents the limitations of using structure-based approaches to engineer prefusion-stabilized S immunogens.

## Results

### Establishing a high-throughput fusion assay for SARS-CoV-2 S

High-throughput assays for measuring protein mutant expression level in human cells have been developed in previous studies by one of our authors using landing pad cells[29–31], which enable one cell to express one mutant, thereby providing a genotype-phenotype linkage[32,33]. Such assays have also been adopted to study the impact of N-terminal domain (NTD) mutations on SARS-CoV-2 S expression[34]. However, there is no similar assay for measuring fusogenicity. Conventional approaches for quantifying fusogenicity often rely on split fluorescent protein systems[35–40], such as the split GFP system that consists of GFP$_{1-10}$ and GFP$_{11}$[41]. For example, when cells that express hACE2 and GFP$_{1-10}$ are mixed with cells expressing SARS-CoV-2 S and GFP$_{11}$, fusion occurs, and the resultant syncytia fluoresce green. In this study, we pioneered an approach by combining this fluorescence-based fusion assay with the use of landing pad cells to establish a high-throughput fusion assay that is compatible with DMS[42].

Specifically, we constructed a DMS library of membrane-bound S that was expressed by HEK293T landing pad cells, such that each cell would encode and express one S mutant. The DMS library contained all possible amino acid mutations from residues 883 to 1034, which covers HR1 (residues 912-984) and CH (residues 985-1034). All S-expressing cells also expressed mNeonGreen2$_{11}$ (mNG2$_{11}$), which belongs to the split monomeric NeonGreen2 system[43]. At the same time, a stable cell line that expressed hACE2 and mNG2$_{1-10}$ was generated (Fig. S1). For the rest of the study, unless otherwise stated, HEK293T landing pad cells that expressed S and mNG2$_{11}$ are abbreviated as "S-expressing cells" and those that expressed hACE2 and mNG2$_{1-10}$ are abbreviated as "hACE2-expressing cells".

When S-expressing cells were mixed with hACE2-expressing cells, S-expressing cells that encoded fusion-competent mutants would fuse with hACE2-expressing cells to form green-fluorescent syncytia

(Fig. 1a, c, see Methods). In contrast, no fusion would occur with S-expressing cells that encoded fusion-incompetent mutants. Subsequently, fluorescence-activated cell sorting (FACS) was performed to separate the unfused cells and green-fluorescent syncytia, both of which were then analyzed by next-generation sequencing. The fusogenicity of each mutant could be quantified by comparing its frequency between the green-fluorescent syncytia sample and the unfused cell sample. In parallel, the expression level of each mutant was measured in a high-throughput manner as described previously[29,34] (see Methods).

Prior to performing the DMS experiments above, the expression of membrane-bound S in HEK293T landing pad cells was verified via flow cytometry analysis using the RBD antibody CC12.3[44] (Fig. 1b). Moreover, the formation of green-fluorescent syncytia due to the fusion of S-expressing cells and hACE2-expressing cells was also verified by microscopy and flow cytometry (Fig. 1c, d, Fig. S2a). We further showed that such fusion can be inhibited by CC40.8, a neutralizing antibody to the stem helix of the S fusion machinery[45], at the highest concentration tested (Fig. S2b). This result confirmed that the fusion of S-expressing cells and hACE2-expressing cells was mediated by the S protein. We optimized the fusion assay to maximize the formation of green-fluorescent syncytia while minimizing the risk of clogging the cell sorter (Fig. S2c–e).

### Identification of fusion-incompetent S mutations with high expression level

From the DMS results, we computed the fusion score and expression score for each of the 2736 missense mutations, 152 nonsense mutations, and 152 silent mutations (see Methods). A higher expression score indicates a higher S expression level. Similarly, a higher fusion score indicates higher fusogenicity. Both expression score and fusion score were normalized such that the average score of silent mutations was 1 and that of nonsense mutations was 0. Three and two biological replicates were performed for the high-throughput expression and fusion assays, respectively. The Pearson correlation coefficient of expression scores among replicates ranged from 0.72 to 0.79, whereas that of fusion scores between replicates was 0.61, confirming the reproducibility of our DMS experiments (Fig. S3a, b). In addition, the expression score distribution and fusion score distribution of silent mutations were significantly different from those of nonsense mutations (Fig. S3c, d), indicating that our DMS experiments could distinguish mutants with different expression and fusogenicity levels. The expression score and fusion score for individual mutations are shown in Fig. S4 and Supplementary Data 1.

Since our fusion assay measured the fusogenicity at the cell level rather than at the single molecule level, the fusion score would be influenced by the expression level even if the fusogenicity per S molecule remained constant. Consistently, the fusion score positively correlated with the expression score (Fig. 2a). To correct for the effect of S expression level on fusogenicity, we computed an adjusted fusion score, which represented the residual of a linear regression model of fusion score on expression score (Fig. 2b). Mutations that had a low adjusted fusion score and a high expression score included the well-known prefusion-stabilizing mutations K986P and V987P that were used in current COVID-19 vaccines[46,47] (Fig. 2b), substantiating that our method could identify prefusion-stabilizing mutations.

Previous studies have shown that the expression of S with K986P/V987P can be improved by additional mutations[24–27], as exemplified by an S construct known as HexaPro, which contains mutations F817P, A892P, A899P, A942P on top of K986P and V987P. Except for F817P, the other mutations in HexaPro were all present in our DMS library. Consistent with the original report of HexaPro[24], our DMS data showed that A899P had minimal influence on S expression, whereas A892P and A942P noticeably increased S expression (Fig. 2a, b). These observations further validated our DMS data.

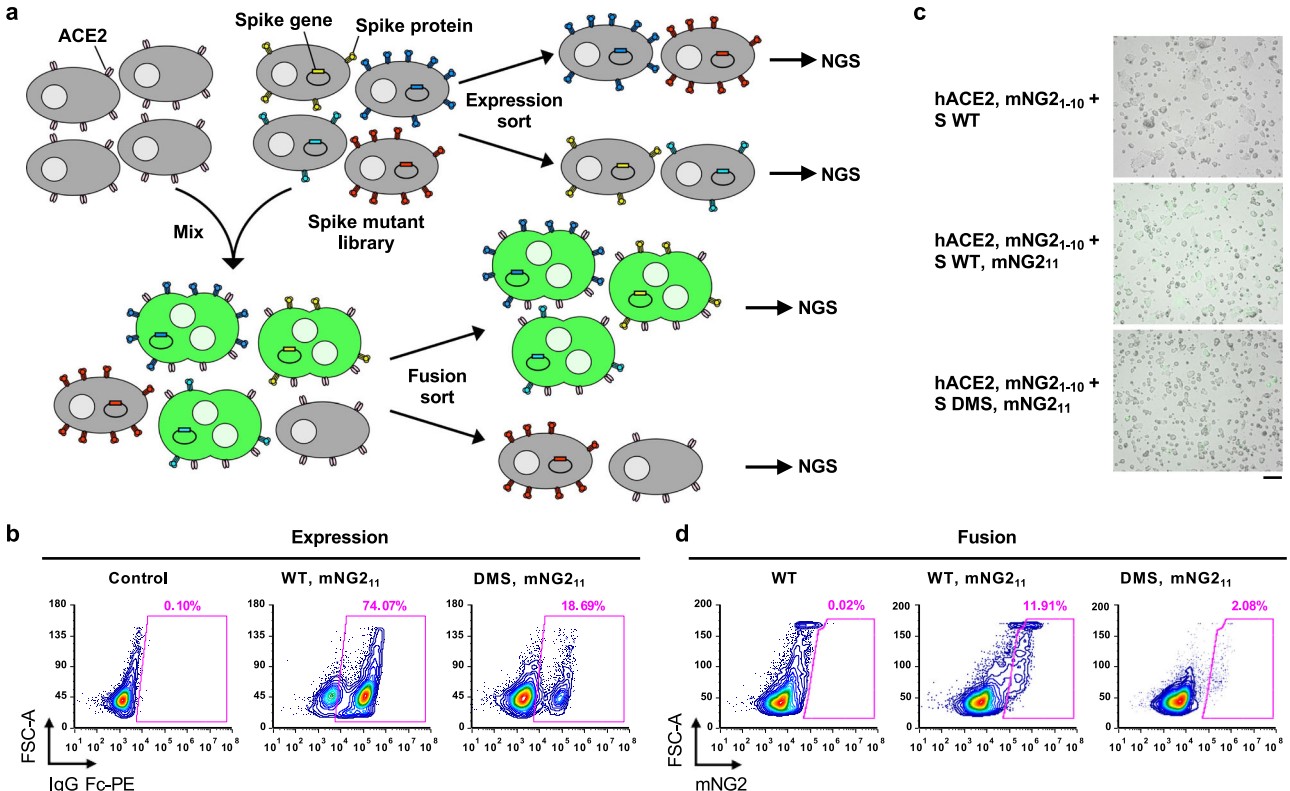

**Fig. 1 | Measuring protein expression and fusogenicity of SARS-CoV-2 S mutations using deep mutational scanning. a** Schematic of high-throughput expression and fusion assays for S mutants. ACE2-expressing and spike-expressing cells also express $mNG2_{1-10}$ and $mNG2_{11}$, respectively. Four-way and two-way sorting were performed for expression and fusion sort, respectively. Fig. S11c, d show gating strategies for sorting. Next-generation sequencing (NGS) was performed on an amplicon that spanned amino acids 883–1034 of S. **b** Flow cytometry analysis of S protein expression in HEK293T landing pad cells that encoded WT S or the DMS library. Primary antibody used was CC12.3, an RBD antibody[44]. **c** Fluorescent micrographs of co-culturing S-expression cells with hACE2-expressing cells. Micrographs are representative of $n = 3$ independent biological experiments. Scale bar: 100 μm. **d** Flow cytometry analysis of fusion activity of co-culturing hACE2-expressing cells with HEK293T landing pad cells that encoded WT S or the DMS library. Components of split mNG2 are indicated where present. Fig. S11b shows gating strategy for flow cytometry analysis of fusion.

## Validation and combinations of prefusion-stabilizing mutations

Besides K986P and V987P, we also identified other mutations in HR1 and CH that had a low adjusted fusion score and a high expression score, particularly T961F, D994E, D994Q and Q1005R (Fig. 2b, c). Of note, D994E and D994Q were at the same residue position and chemically similar. By expressing these four mutations individually using HEK293T landing pad cells, we validated that they indeed improved the surface expression of S (Figs. 3a, S5a) and prevented the formation of syncytia when incubated with hACE2-expressing cells (Figs. 3d, S6a, b). Consistent with the DMS data (Fig. 2), the effects of T961F, D994E, D994Q and Q1005R on S expression and fusogenicity were comparable to K986P and V987P in the validation experiments. As a control, we also selected two mutations that had a high adjusted fusion score and a high expression score, namely S943H and A944S (Fig. 2b), and validated their enhancement in S expression and fusogenicity (Figs. 3b, e, S5b, S6c, d).

Subsequently, we combined the validated fusion-incompetent mutations K986P, V987P, D994Q and Q1005R to generate double (K986P/V987P: '2P'), triple (K986P/V987P/D994Q: '2PQ', K986P/V987P/Q1005R: '2PR') and quadruple (K986P/V987P/D994Q/Q1005R: '2PQR') mutants of membrane-bound S. Surface expression of these mutation combinations was higher than that of WT, but comparable with each other (Figs. 3c, S5c). As expected, none of these S mutation combinations fused with hACE2-expressing cells (Figs. 3f, S6e, f). We further tested the expression of soluble S ectodomain with different mutants. Interestingly, addition of the D994Q to 2P improved expression of soluble S ectodomain by approximately three-fold while

the Q1005R drastically reduced expression of soluble S (Fig. S7). Q1005R seemed to increase the formation of higher order oligomers of soluble S ectodomain, as observed by a peak higher than the expected size of trimeric S ectodomain in size exclusion chromatography of all mutants that contained Q1005R (Fig. S7b). These observations indicate that certain mutations can improve the expression level of S in membrane-bound form but not soluble ectodomain form.

## Structural and biophysical characterization of 2PQ spike

Due to the improvement of 2PQ over 2P in soluble S ectodomain expression, we proceeded with biophysical characterization of 2PQ to rationalize the prefusion-stabilization mechanism of D994Q. The prefusion conformation of 2PQ was confirmed by low-resolution cryogenic electron microscopy (Figs. 4a, b, S8a). While the structure could not be resolved at atomic resolution, the result allowed us to confirm that the 2PQ mutant is in the prefusion-stabilized conformation. In addition, the electron density of the protein backbone clearly showed that the helix containing residue 944 is shifted towards the helix containing residue Q755 (Fig. S8b). This observation is corroborated by in silico mutagenesis using Rosetta, which showed that the helices are brought together in proximity so that D994Q forms an intraprotomer hydrogen bond with Q758 to stabilize the prefusion conformation (Fig. 4d). Differential scanning fluorimetry revealed that both 2P and 2PQ had an apparent melting temperature at approximately 46.5 °C, similar to the previously reported value for 2P[24]. Nevertheless, 2PQ had another peak at approximately 62 °C,

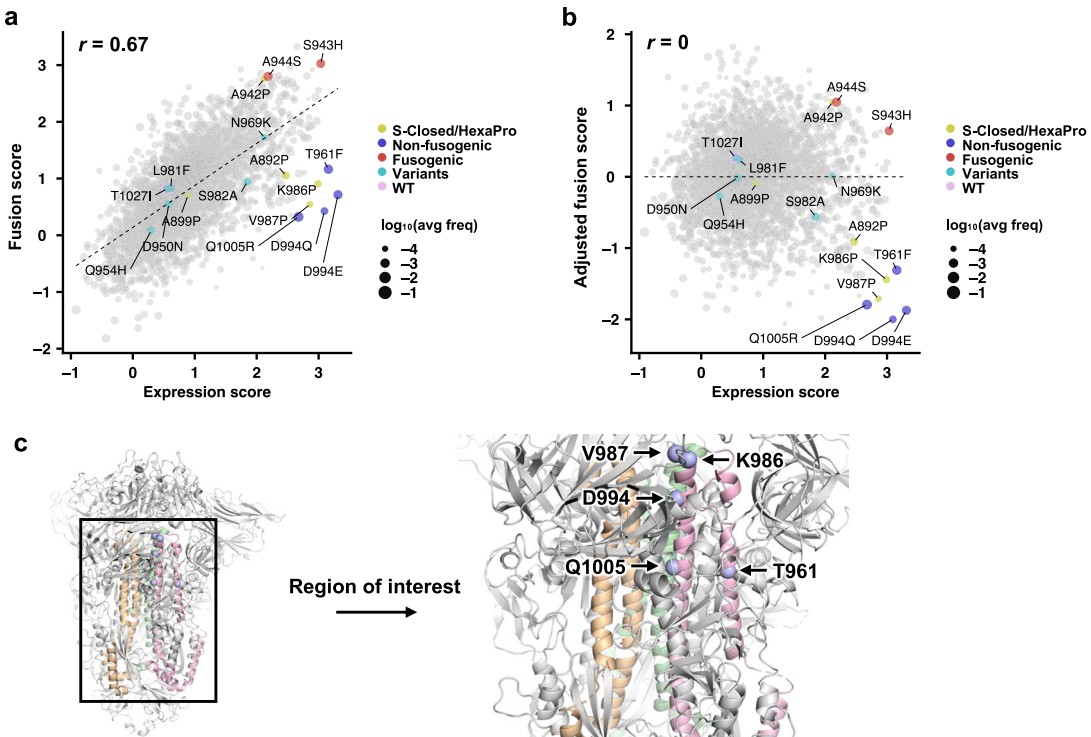

**Fig. 2 | Expression and fusion scores of individual mutations in the DMS library.** **a** Plot of fusion score against expression score for each mutant is shown. WT is indicated in pink. Mutations used in S-Closed[25] and/or HexaPro[24] are in yellow. Representative fusion-incompetent mutations identified in this study are in purple (non-fusogenic). Representative mutations that enhance S fusogenicity are in red (fusogenic). Mutations found in major SARS-CoV-2 variants (Supplementary Data 1) are in teal (variants). Each data point represents one mutation in the DMS library. Individual data points are sized according to average frequency of the corresponding mutations. Pearson correlation coefficient, $r$, is shown. **b** Plot of adjusted fusion score against expression score for each mutant is shown. Pearson correlation coefficient, $r$, is shown. **c** Locations of fusion-incompetent mutations are indicated by light blue spheres. Regions that are mutated in the DMS library are colored wheat, green and pink for each monomer. Other regions on the S are colored in grey. Data are from Supplementary Data 1.

suggesting that the additional D994Q mutation prevents immediate, complete unfolding of S (Fig. 4c). The stabilizing effect of D994Q, however, was not as pronounced as a combination of F817P, A892P, A899P, A942P that were used in HexaPro, which not only showed two peaks in the differential scanning fluorimetry analysis, but also shifted the first apparent melting temperature by 5 °C[24].

Finally, we tested whether D994Q altered the antigenicity of the S protein. We compared the binding of 2P and 2PQ to various S antibodies, including CC12.3 (RBD)[44], S2M28 (NTD)[48], CC40.8 (S2 stem helix)[45], and COVA1-07 (S2 HR1)[49], using flow cytometry. 2P and 2PQ showed similar binding affinity to CC12.3 and S2M28 (Figs. 4e, S9a, b). However, when assayed for binding with COVA1-07 or with CC40.8, 2PQ had weaker binding than 2P (Figs. 4e, S9c, d). Of note, COVA1-07 only binds efficiently when S is in an open conformation that has transitioned away from the prefusion conformation[49]. Similarly, the binding of CC40.8 to S requires partial disruption of the prefusion S trimer and is shown to be weakened by prefusion-stabilizing mutations[45]. Therefore, our result substantiates that D994Q can further enhance the prefusion stability of 2P, which is known to insufficiently stabilize the prefusion conformation[24,25,50]. Collectively, these data reveal a prefusion-stabilization mechanism of D994Q and demonstrate its minimal impact on the antigenicity of the head domain of S. Future studies should explore whether D994Q or other prefusion-stabilizing mutations identified in this study can further improve the stability of other prefusion-stabilized S constructs, such as S-Closed[25], HexaPro[24], and VFLIP[27], while retaining its antigenicity.

## Discussion
Structure-based design[28] of prefusion-stabilized class I viral fusion proteins has been successfully applied to HIV[51–54], RSV[55], Nipah[56], Lassa[57], Ebola[58], and more recently SARS-CoV-2[24–27]. Although structure-based design is an effective approach for prefusion-stabilization, it requires structural determination and subsequent expression, purification, and characterization of each candidate mutation individually. This laborious experimental process limits the comprehensiveness of using a structure-based approach to identify prefusion-stabilizing mutations. In this study, we established a high-throughput approach to measure the fusogenicity of thousands of mutations in parallel. This approach enables systematic identification of prefusion-stabilizing mutations without relying on structural information. While we only provide a proof-of-concept using the SARS-CoV-2 S protein, our approach can be adopted to fusion proteins of other viruses with known cell surface receptors. Given that prefusion-stabilization is critical for viral immunogen design[28,59], our work here should advance the process of viral vaccine development.

One interesting finding in this study is that the expression of membrane-bound (i.e. full-length) S protein does not necessarily correlate with the expression of soluble S ectodomain, as exemplified by Q1005R. In addition, our results show that S943G, A944G and A944P mutations, which have been shown to increase the expression of soluble S ectodomain[25], do not increase the expression of membrane-bound S protein. These observations indicate that the ectodomain of the S protein has some long-range interactions with its native transmembrane domain. As a result, caution is needed when extrapolating the results obtained from full-length S protein to soluble S ectodomain, or vice versa. However, since most COVID-19 vaccines on the market are based on the full-length membrane-bound S protein[60], the results from our high-throughput fusion and expression assays, which are also based on full-length membrane-bound S protein, are directly applicable to COVID-19 vaccine development.

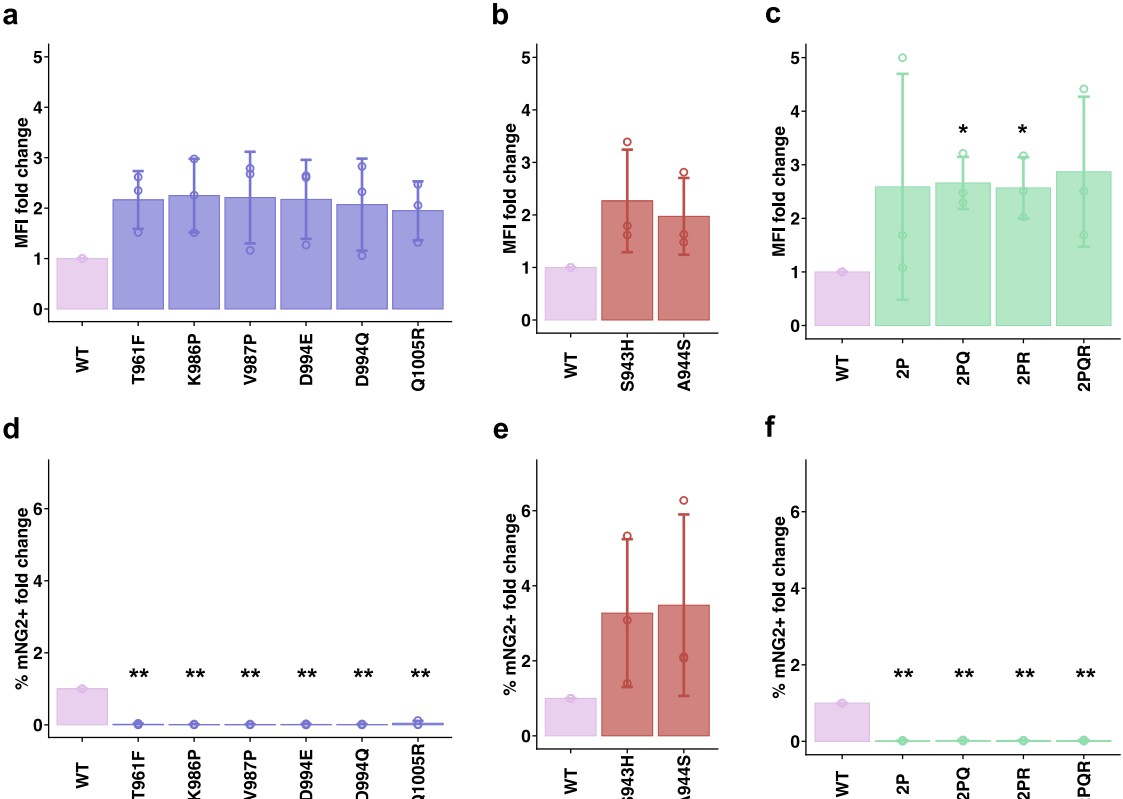

**Fig. 3 | Validation of candidate prefusion-stabilizing mutations. a–c** Expression of prefusion-stabilizing mutations (**a**), fusion-enhancing mutations (**b**), and combinations of candidate prefusion-stabilizing mutations of S (**c**) relative to WT. Of note, the numerical values of fold change in median fluorescence intensity (MFI) indicate relative and not absolute fold changes in surface expression levels of S. **d–f** Fold change in fusion activity of candidate prefusion-stabilizing mutations (**d**), fusion-enhancing mutations (**e**), and combinations of candidate prefusion-stabilizing mutations of S (**f**) relative to WT at 3 h post-mixing with hACE2-expressing cells. Abbreviations for combinatorial mutations are as follows: 2P, K986P/V987P; 2PQ, K986P/V987P/D994Q; 2PR, K986P/V987P/Q1005R; 2PQR, K986P/V987P/D994Q/Q1005R. Data are from $n = 3$ independent replicates, and shown as mean ± standard deviation. $p$-values were calculated from a two-sided Welch's $t$-test and are listed in the Source Data file; * $p < 0.05$; ** $p < 0.01$. Source data are provided as a Source Data file.

Although most SARS-CoV-2 neutralizing antibodies target RBD[61], recent studies have shown that antibodies to S2 can also neutralize, albeit often at a lower potency[45,62–65]. As a result, understanding the evolutionary constraints of S2 is relevant to SARS-CoV-2 antigenic drift and to design of more universal coronavirus vaccines. While many mutations in HR1 and CH, including those of major SARS-CoV-2 variants (Table S1), do not negatively impact the expression or fusogenicity of the S protein (Fig. 2b), HR1 and CH show high degrees of evolutionary conservation among betacoronaviruses (Fig. S10). This observation could be due to low levels of positive selection pressure on HR1 and CH, since most neutralizing antibodies are directed towards the RBD[61]. Alternatively, besides S protein expression and fusogenicity, other evolutionary constraints on HR1 and CH may be present in vivo. Future studies of the relationship among S protein expression, fusogenicity, and virus replication fitness will provide important biophysical insights into the evolution of SARS-CoV-2.

Since RBD is present in the prefusion conformation but not the postfusion conformation[5–7] and is the major target of neutralizing antibodies[61], this study used an RBD antibody to probe for surface expression. Nevertheless, we acknowledge that the folding of prefusion S can be more comprehensively probed by antibodies to conformational epitopes in the NTD and S2 subunit. Furthermore, due to the technical difficulties in sorting large syncytia, co-culturing of S-expressing cells and hACE2-expressing cells could only be performed for relatively short durations before sorting, leading to fewer syncytia and potentially lower reproducibility of results across replicates. Alternative strategies including microfluidics-based fusion

experiments can be explored to obviate the kinetic limitations of the current fusion assay.

If the prefusion-stabilizing mutations of betacoronavirus S protein were not reported in late 2010s[17,66], it would not have been possible to develop an effective COVID-19 vaccine at the speed that occurred, even with the availability and utilization of the mRNA vaccine technology. It is unclear whether the next pandemic will be caused by a virus that we have prior knowledge about. Consequently, while the speed of vaccine manufacturing has been revolutionized by the mRNA vaccine technology[67], it is equally important to maximize the speed of immunogen design so that we are fully prepared for the next pandemic. We believe our work here provides an important step in that regard.

## Methods
### Cell culture
Human embryonic kidney 293T (HEK293T) landing pad cells obtained from Dr. Kenneth A. Matreyek (Case Western Reserve University) were grown and maintained in complete growth medium: Dulbecco's modified Eagle medium (DMEM) with high glucose (Gibco), supplemented with 10% v/v fetal bovine serum (FBS; VWR), 1× non-essential amino acids (Gibco), 100 U/mL penicillin and 100 µg/mL streptomycin (Gibco), 1× GlutaMAX (Gibco) and 2 µg/mL doxycycline (Thermo Scientific) at 37 °C, 5% $CO_2$ and 95% humidity. Expi293F cells (Gibco, catalog number A14527) were grown and maintained in Expi293 expression medium (Gibco) at 37 °C, 8% $CO_2$, 95% humidity and 125 rpm according to the manufacturer's instructions.

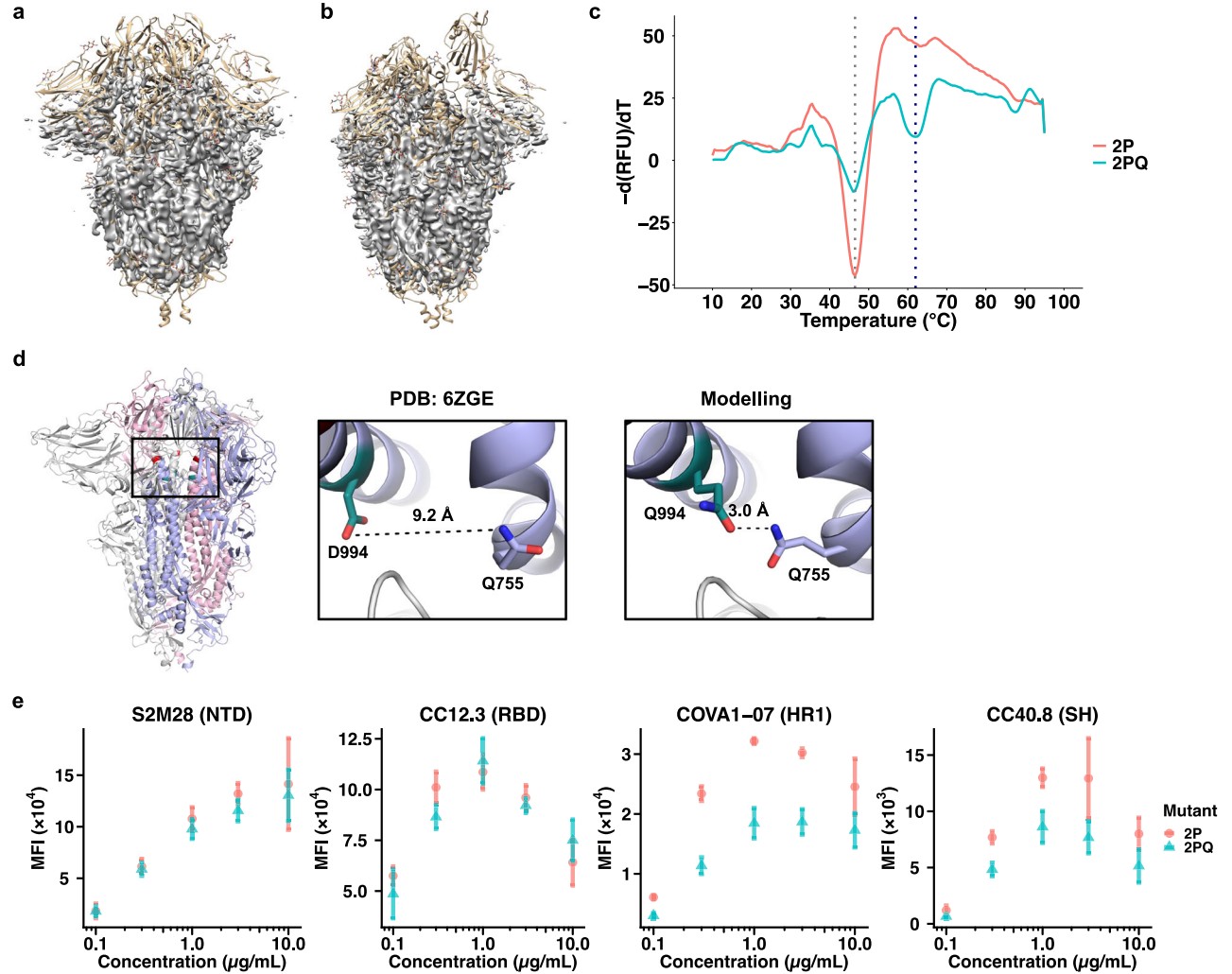

**Fig. 4 | Biophysical characterization of 2PQ spike. a, b** EM density map (colored grey) of 2PQ fitted on S with all-down RBD (PDB: 6VXX) (**a**), and one-up RBD (PDB: 6VYB) (**b**). **c** The first differential curves for the relative fluorescence unit (RFU) from differential scanning fluorimetry with respect to temperature are shown for soluble 2PQ and 2P. Grey dotted line indicates the first apparent melting temperature ($T_m$) of 2P and 2PQ at approximately 46.5 °C; blue dotted line indicates the second apparent $T_m$ of 2PQ at approximately 62 °C. **d** D994Q allows formation of an additional intraprotomer hydrogen bond as shown by structural modelling.

Distinct protomers are in grey, light blue and pink. The Q758 and Q994 side chains are shown as sticks representation. Hydrogen bond is indicated in black dashed line with the distance indicated. **e** Titration of S bearing 2P or 2PQ mutations with an N-terminal domain (NTD), receptor-binding domain (RBD), first heptad repeat (HR1), or stem helix (SH) antibody via flow cytometry. Median fluorescence intensities (MFI) are shown as mean ± standard deviation from *n* = 3 independent replicates. Source data are provided as a Source Data file.

## Landing pad plasmids

attB plasmids each encoding (hACE2, an internal ribosomal entry site [IRES], and hygromycin resistance: attB-hACE2), (hACE2, an IRES, general control nondepressible 4 [GCN4] leucine zipper fused to mNG2$_{1\text{-}10}$, a (GSG) P2A self-cleaving peptide, and hygromycin resistance: attB-hACE2-mNG2-1-10), and (S with the PRRA motif in the furin cleavage site deleted, an IRES, GCN4 leucine zipper fused to mNG2$_{11}$, a (GSG) P2A self-cleaving peptide, and puromycin resistance: attB-S-mNG2-11) were constructed and assembled via polymerase chain reaction (PCR). The sequence of S used in this study was the ancestral (Wuhan-Hu-1) strain (GenBank accession ID: MN908947.3)[68]. The PRRA motif in the furin cleavage site was deleted to prevent spontaneous fusion of S-expressing cells with each other[69]. For experimental validation, mutants of S were individually constructed using PCR-based site-directed mutagenesis. Pairs of primers used for PCR-based site directed mutagenesis are listed in Table S2.

## Deep mutational scanning library construction

Cassette primers for DMS library construction are listed in Table S3. Cassette primers were resuspended in MilliQ $H_2O$ such that the final concentration of all primers is 10 µM. Forward cassette primers, named as CassetteX_N (X = 1, 2, …, 19; N = 1, 2, …, 8), that belong to the same cassette (i.e., the same value of X) were mixed in equimolar ratios. Each forward cassette primer also carried unique silent mutations (i.e. synonymous mutations) to help distinguish between sequencing errors and true mutations in downstream sequencing data analysis as described previously[70]. For the first round of PCR, two sets of reactions were set up. The first set had the mixed cassette primers and 5′-ACG ACG TCT CCT TCT CTA GGA AAG TGG GCT TTG C-3′ as forward and reverse primers, respectively. The second set had 5′-TGC TCG TCT CCA AAG TGA CAC TGG CCG ACG CCG G-3′ and CassetteX_Rprimers (X = 1, 2, …, 19) as forward and reverse primers, respectively. Since we had 19 cassettes, there were 19 PCRs for each of the two sets of reactions. For both sets, the template used was attB-S-mNG2-11.

Thereafter, products corresponding to the correct size were excised and purified using Monarch DNA Gel Extraction kit (NEB). For the second round of PCR, 10 ng of PCR product from each of the first and second sets in the same cassette were mixed. 5'-ACG ACG TCT CCT TCT CTA GGA AAG TGG GCT TTG C-3' and 5'-TGC TCG TCT CCA AAG TGA CAC TGG CCG ACG CCG G-3' were used as the forward and reverse primers, respectively. PCR products corresponding to the correct size were excised and purified using DNA Gel Extraction kit (NEB). 100 ng of each gel-purified PCR products (total of 19) were mixed and digested with BsmBI restriction enzyme (NEB) for 2 h at 55 °C. Then, the product was purified using PureLink PCR Purification kit (Invitrogen) and served as the insert.

To amplify the vector, attB-S-mNG2-11, 5'-CAC TCG TCT CGA GAA GGC GTG TTC GTG TCC AAC G-3', and 5'-GGC CCG TCT CAC TTT GTT GAA CAG CAG GTC CTC G-3' were used as template, forward primer, and reverse primer, respectively. The PCR product was digested with DpnI (NEB) for 2 h at 37 °C, purified with PureLink PCR Purification kit (Invitrogen), digested with BsmBI restriction enzyme (NEB) for 2 h at 55 °C, and purified again using a PureLink PCR Purification kit (Invitrogen). All PCRs were performed using PrimeSTAR Max DNA Polymerase (Takara) according to the manufacturer's instructions.

BsmBI-digested vector and insert were ligated in a molar ratio of 1:100 to a total of 1 μg using T4 DNA ligase (NEB) for 2 h at room temperature. A control ligation reaction was set up by only having the BsmBI-digested vector (no insert). 1 μL ligation reaction products were transformed into chemically competent DH5α *Escherichia coli* cells and plated onto agar plates with 100 μg/mL ampicillin. The ligation mixture that contained vector and insert had at least 10 times more colonies than the control reaction. Subsequently, the ligation mixture was column-purified using a PureLink PCR Purification kit and eluted in 10 μL of MilliQ H$_2$O. 1 μL of the purified ligated product was mixed with 30 μL MegaX DH10β T1$^R$ electrocompetent *E. coli* cells (NEB) into an electroporation cuvette with a 1 mm gap (BTX). Electroporation was performed at 2.0 kV, 200 Ω and 25 μF using an ECM 830 square wave electroporation system (BTX). 1 mL of SOC recovery medium (NEB) was added immediately into cells after electroporation. Two electroporation reactions were performed. Cells were recovered for 1 h at 37 °C with shaking at 250 rpm. After recovery, cells were collected via centrifugation, resuspended in 400 μL lysogeny broth (LB), plated onto 150 mm agar plates supplemented with 100 μg/mL ampicillin, and incubated overnight at 37 °C. At least $1 \times 10^6$ colonies were scrape-harvested with LB broth and plasmids were extracted using a PureLink Plasmid Midiprep kit (Invitrogen).

## Landing pad cell transfection

$6.0 \times 10^5$ HEK293T landing pad cells in 1.35 mL of complete growth medium were seeded per well of a 6-well plate. 1.7 μg of attB-hACE2-mNG2-1-10 plasmid or attB-S-mNG2-11 plasmid were added into 5 μL FuGENE 6 transfection reagent (Promega) and OptiMEM (Gibco) to a total volume of 240 μL. The transfection mixture was subsequently added dropwise into cells. Transfection was carried out on the same day as seeding. One day post-transfection, 500 μL of complete growth medium was added to cells. Three days post-transfection, medium was discarded, cells were washed with 1× PBS, and incubated in negative selection medium (complete growth medium supplemented with 10 nM AP1903) for one day at 37 °C, 5% CO$_2$ and 95% humidity. Then, the medium was discarded, cells were washed with 1× PBS, and recovered in complete growth medium for two days at 37 °C, 5% CO$_2$ and 95% humidity. Cells were then trypsinized and grown in positive selection medium indefinitely: hACE2- and S-expressing cells were maintained in hygromycin medium (complete growth medium supplemented with 100 μg/mL hygromycin B [Invivogen]) and puromycin medium (complete growth medium supplemented with 1 μg/mL puromycin [Invivogen]), respectively.

To construct the S2 HR1/CH DMS cell line, the above protocol was used with modifications: $3.5 \times 10^6$ cells in 8 mL of complete growth medium in a T75 flask were transfected with 7.1 μg of the DMS plasmid library and 29 μL of FuGENE6 transfection reagent in 1.4 mL of Opti-MEM. For positive selection and regular maintenance, puromycin medium was used.

## Flow cytometry

To validate hACE2 surface expression after transfection, landing pad cells were harvested via centrifugation at $300 \times g$ for 5 min at 4 °C, resuspended in ice-cold FACS buffer (2% v/v FBS, 50 mM EDTA in DMEM supplemented with high glucose, L-glutamine and HEPES, without phenol red [Gibco]), and incubated with 2 μg/mL of SARS-CoV-2 S RBD-IgG Fc for 1 h at 4 °C. Then, cells were washed once, and resuspended with ice-cold FACS buffer. Cells were incubated with 1 μg/mL of phycoerythrin (PE)-conjugated anti-human IgG Fc (Bio-Legend, clone M1310G05, catalog number 410708). Cells were washed once and resuspended in ice-cold FACS buffer. Cells were analyzed using an Accuri C6 flow cytometer (BD Biosciences). Data was collected using BD Accuri C6 software v264 (BD Biosciences).

The above protocol for verification and quantification of S surface expression was used except cells were incubated with 5 μg/mL of CC12.3[44], an RBD antibody, instead of SARS-CoV-2 S RBD-IgG Fc, for 1 h at 4 °C. To quantify fold change in surface expression of S relative to WT based on median fluorescence intensity (MFI), Eq. (1) was used in the plot of FSC-A against PE:

$$\text{MFI}_{\text{FC}} = \frac{\text{MFI}_{\text{mutant}} - \text{MFI}_{\text{control}}}{\text{MFI}_{\text{WT}} - \text{MFI}_{\text{control}}} \quad (1)$$

MFI values were obtained after plotting data in FCS Express Flow Cytometry software v6 (De Novo Software). Gating strategy is shown in Fig. S11a.

To assess fusogenicity of S (WT or mutants), an equal number of hACE2, mNG2$_{1-10}$- and S, mNG2$_{11}$-expressing cells were mixed such that the total cell number is $5.0 \times 10^5$ cells per mL of complete growth medium. Cells were co-cultured for 3 h at 37 °C, 5% CO$_2$ and 95% humidity. Cells were then harvested and resuspended in ice-cold FACS buffer. Cells were analyzed using an Accuri C6 flow cytometer (BD Biosciences). Data was collected using BD Accuri C6 software v264 (BD Biosciences). Gating strategy is shown in Fig. S11b. The percentage of mNG2-positive events of mutants relative to that of WT S was calculated.

For titration of S-2P or S-2PQ, HEK293T landing pad cells stably expressing membrane-bound S-2P or S-2PQ were incubated with 0, 0.1, 0.3, 1.0, 3.0 or 10.0 μg/mL of S2M28, CC12.3, COVA1-07, or CC40.8 antibody for 1 h in ice-cold FACS buffer. Cells were washed and then incubated with 1 μg/mL PE-conjugated anti-human IgG Fc for 1 h at 4 °C. Cells were washed, resuspended in ice-cold FACS buffer, and analyzed for levels of PE using an Accuri C6 flow cytometer. Gating strategy is shown in Fig. S11a. MFI values were subtracted from those of negative control (0 μg/mL of antibody) and plotted against antibody concentration (Figs. 4e, S10).

## Expression sorting

Cells expressing the S2 HR1/CH DMS library of S were harvested via centrifugation at $300 \times g$ for 5 min at 4 °C. Supernatant was discarded, and cells were resuspended in ice-cold FACS buffer. Cells were incubated with 5 μg/mL of CC12.3 for 1 h at 4 °C. Then, cells were washed once, and resuspended with ice-cold FACS buffer. Cells were incubated with 2 μg/mL of PE anti-human IgG Fc. Cells were washed once, resuspended in ice-cold FACS buffer, and filtered through a 40 μm strainer. Cells were sorted via a four-way sort using a BD FACS Aria II cell sorter and BD FACS Diva software v8.0 (BD Biosciences), or a BigFoot spectral cell sorter and Sasquatch software firmware v888

(Invitrogen) according to PE fluorescence at 4 °C. Cells expressing the highest PE fluorescence were sorted into "bin 3", then the next highest into "bin 2", followed by "bin 1" and then "bin 0". Each bin had ~25% of the singlet population. Gating strategy is shown in Fig. S11c. Number of cells collected per bin per replicate is shown in Table S4. Of note, since CC12.3 binds to the RBD[44], an independently folded region of S that is present only in the prefusion but not postfusion conformation[1,71], our sort was based on the expression of prefusion S.

## Fusion sorting

Cells expressing the HR1/CH DMS library of S, and cells expressing hACE2 were resuspended in complete growth medium and filtered through a 40 μm cell strainer to obtain single cell suspensions. $2.5 \times 10^6$ cells of each were mixed in a T-75 flask and complete growth medium added to a total volume of 10 mL. Six co-cultures were set up, with one of the co-cultures acting as a negative, non-fluorescent control by mixing hACE2- and S-expressing cells that do not have split mNG2. Co-cultures were incubated for 3 h at 37 °C, 5% $CO_2$ and 95% humidity. Subsequently, cells were harvested and pelleted via centrifugation at $300 \times g$ for 5 min at 4 °C. Supernatant was discarded, and cells were resuspended in ice-cold FACS buffer. Cells were sorted via a two-way sort using a BigFoot spectral cell sorter (Invitrogen) according to presence or absence of mNG2 fluorescence at 4 °C. Gating strategy is shown in Fig. S11d. Number of cells collected per bin per replicate is shown in Table S5.

## Post-sorting genomic DNA extraction

After FACS, cell pellets were obtained via centrifugation at $300 \times g$ for 15 min at 4 °C, and the supernatant was discarded. Genomic DNA was extracted using a DNeasy Blood and Tissue Kit (Qiagen) following the manufacturer's instructions with a modification: resuspended cells were incubated and lysed at 56 °C for 30 min instead of 10 min.

## Deep sequencing

After genomic DNA extraction, the region of interest was amplified via PCR using 5′-CAC TCT TTC CCT ACA CGA CGC TCT TCC GAT CTA CAT CTG CCC TGC TGG CCG GCA CA-3′ and 5′-GAC TGG AGT TCA GAC GTG TGC TCT TCC GAT CTG CAA AAG TCC ACT CTC TTG TTG TC-3′ as forward and reverse primers, respectively. A maximum of 500 ng of genomic DNA per 50 μL PCR reaction was used as template; 4 μg of genomic DNA per expression or fusion bin, per replicate, was used as template. PCR was performed using KOD DNA polymerase (Takara) with the following settings: 95 °C for 2 min, 25 cycles of (95 °C for 20 s, 56 °C for 15 s, 68 °C for 20 s), 68 °C for 2 min, 12 °C indefinitely. All eight 50 μL reactions per bin per replicate were mixed after PCR. 100 μL of product per bin per replicate was used for purification using a PureLink PCR Purification kit. Subsequently, 10 ng of the purified PCR product per bin per replicate was appended with Illumina deep sequencing barcodes via PCR using KOD DNA polymerase with the following settings: 95 °C for 2 min, 9 cycles of (95 °C for 25 s, 56 °C for 15 s, 68 °C for 20 s), 68 °C for 2 min, 12 °C indefinitely. Barcoded products were mixed and sequenced with a MiSeq PE300 v3 flow cell (Illumina).

## Analysis of deep sequencing data

Forward and reverse reads were merged via PEAR[72]. Using custom Python code, the merged reads were translated and matched to the corresponding mutant. Counts for expression and fusion bins for each replicate were tabulated. For each replicate, the frequency of each mutant was calculated as the count of that mutant divided by the total number of counts in that bin, as shown in Eq. (2):

$$F_{mut, binX} = \frac{C_{mut,binX}}{\Sigma C_{binX}} \text{ for X} = 0, 1, 2, 3, \text{mNG2}^-, \text{mNG2}^+ \quad (2)$$

For each replicate, the weighted expression score for each mutant ($W_{mut}$) was calculated using Eq. (3):

$$W_{mut} = \frac{(F_{mut,bin0} \times 0.25) + (F_{mut,bin1} \times 0.5) + (F_{mut,bin2} \times 0.75) + (F_{mut,bin3} \times 1)}{F_{mut,bin0} + F_{mut,bin1} + F_{mut,bin2} + F_{mut,bin3}}$$

$$(3)$$

The weighted expression scores were normalized ($W_{mut}^{norm}$) such that the average $W_{mut}$ of nonsense mutations equals 0, and the average $W_{mut}$ of silent mutations equals 1 using Eq. (4):

$$W_{mut}^{norm} = \frac{W_{mut} - W_{nonsense}^{avg}}{W_{silent}^{avg} - W_{nonsense}^{avg}} \quad (4)$$

The final expression score ($W_{mut}^{avg}$) for each mutant was calculated using Eq. (5):

$$W_{mut}^{avg} = \frac{1}{3} \times \left( W_{mut}^{norm,rep1} + W_{mut}^{norm,rep2} + W_{mut}^{norm,rep3} \right) \quad (5)$$

Fusion scores ($U_{mut}$) were calculated for each replicate by the formula shown in Eq. (6):

$$U_{mut} = \log_{10} \left( \frac{F_{mut,mNG2^+}}{F_{mut,mNG2^-}} \right) \quad (6)$$

Fusion scores were normalized ($U_{mut}^{norm}$) such that the $U_{mut}^{avg}$ of silent mutations equals 1, and the $U_{mut}^{avg}$ of nonsense mutations equals 0 using Eq. (7):

$$U_{mut}^{norm} = \frac{U_{mut} - U_{nonsense}^{avg}}{U_{WT}^{avg} - U_{nonsense}^{avg}} \quad (7)$$

Then, the final average score ($U_{mut}^{avg}$) for each mutant was calculated using Eq. (8):

$$U_{mut}^{avg} = \frac{1}{2} \times \left( U_{mut}^{norm,rep1} + U_{mut}^{norm,rep2} \right) \quad (8)$$

Adjusted fusion score of each mutant is equal to the residual of that mutant in a linear regression model of fusion score against expression score. The linear regression model and residuals were calculated using the 'lm' and 'resid' functions in RStudio v2022.12.0+353.

## Sequence conservation analysis

Sequences were obtained from GenBank or GISAID (Tables S1, S7). A BLAST database was created, and the reference sequence of the DMS region (residues 883-1034) was used to run tblastn to generate BlastXML files. The reference sequence used was the founder strain of SARS-CoV-2 (GenBank accession number: MN908947.3)[68]. Extracted information was obtained by running 'XML_Extraction.py'[73]. Multiple alignment using MAFFT was then performed[74]. Sequence conservation was calculated based on the residue conservation at each position relative to the reference sequence. Mean expression score and mean fusion score were calculated by taking the average of the expression scores and fusion scores of all mutants, respectively, at that position.

## Fluorescence microscopy

Images were captured with an ECHO Revolve epifluorescence microscope (ECHO) with a UPLANFL N 10×/0.30 NA objective (Olympus) using the FITC channel for mNG2 fluorescence. Brightfield images were also obtained using white light. Fluorescent and brightfield images were then overlaid. Identical exposure and intensity settings were used to capture images. Scale bars correspond to 100 μm for all micrographs.

## Cryogenic electron microscopy

To prepare cryoEM grid, an aliquot of 3.5 μL purified protein at ~1 mg/mL concentration was applied to a 300-mesh Quantifoil R1.2/1.3 Cu grid pre-treated with glow-discharge, blotted in a Vitrobot Mark IV machine (force -5, time 3 s), and plunge-frozen in liquid ethane. The grid was loaded in a Titan Krios microscope equipped with Gatan BioQuantum K3 imaging filter and camera. A 10-eV slit was used for the filter. Data collection was done with serialEM v4.0[75]. Images were recorded at 130,000× magnification, corresponding to a pixel size of 0.33 Å/pix at super-resolution mode of the camera. A defocus range of -0.8 μm to -1.5 μm was set. A total dose of 50 e⁻/Å² of each exposure was fractionated into 50 frames. The first two frames of the movie stacks were not included in motion-correction. CryoEM data processing was performed on the fly with cryoSPARC Live v3.3.2 (Structura Biotechnology)[76] following regular single-particle procedures. Statistics are provided in Table S8. Structures were visualized using UCSF ChimeraX v1.5 (UCSF).

## Rosetta-based mutagenesis

The structure of S was obtained from the Protein Data Bank (PDB ID: 6ZGE). N-acetyl-D-glucosamine and water molecules were removed using PyMOL v2.4.0 (Schrödinger), and amino acids were renumbered using pdb-tools[77]. The 'fixbb' application in Rosetta v3.11 (Rosetta-Commons) was used to generate the D994Q mutation in all protomers. One-hundred poses were obtained, and the lowest scoring pose was used for further processing. A constraint file was generated using the lowest-scoring pose from fixed backbone mutagenesis as input, and the 'minimize_with_cst' application in Rosetta. Fast relax was subsequently performed using the 'relax' application[78] with the constraint file. The lowest scoring pose out of thirty was used for structural analysis.

## Antibody expression and purification

Codon-optimized oligonucleotides encoding the heavy chain and light chain of the indicated antibodies were cloned into phCMV3 plasmids in an IgG1 Fc format with a mouse immunoglobulin kappa signal peptide. Plasmids encoding the heavy chain and light chain of antibodies were transfected into Expi293F cells using an Expifectamine 293 transfection kit (Gibco) in a 2:1 mass ratio following the manufacturer's protocol. Supernatant was harvested 6 days post-transfection and centrifuged at 4000 × g for 30 min at 4 °C to remove cells and debris. The supernatant was subsequently clarified using a polyethersulfone membrane filter with a 0.22 μm pore size (Millipore).

CaptureSelect CH1-XL beads (Thermo Scientific) were washed with MilliQ $H_2O$ thrice, and resuspended in 1× PBS. The clarified supernatant was incubated with washed beads overnight at 4 °C with gentle rocking. Then, flowthrough was collected, and beads washed once with 1× PBS. Beads were incubated in 60 mM sodium acetate, pH 3.7 for 10 min at 4 °C. The eluate containing antibody was buffer-exchanged into 1× PBS using a centrifugal filter unit with a 30 kDa molecular weight cut-off (Millipore) four times. Antibodies were stored at 4 °C.

## Soluble S protein expression and purification

SARS-CoV-2 S ectodomain (residues 1-1213, which includes the native signal peptide) with the PRRA motif in the furin cleavage site deleted, C-terminal SGGGG linker, biotinylation site, thrombin cleavage site, Foldon trimerization sequence, and 6×His-tag were all cloned in-frame into a phCMV3 vector via PCR. Site-directed mutagenesis via PCR was performed to generate the indicated mutants of soluble S protein.

Expi293F cells were transfected with vectors encoding the indicated soluble spike protein mutant using an Expifectamine 293 transfection kit following the manufacturer's protocol. Cells were harvested six days post-transfection. The supernatant was collected via centrifugation at 4000 × g for 30 min at 4 °C, and further clarified using a polyethersulfone membrane with a 0.22 μm pore size (Millipore). The clarified supernatant was incubated with washed Ni sepharose excel His-tagged protein purification resin (Cytiva) with gentle rocking overnight at 4 °C. Flow-through was collected. Beads were washed once with 20 mM imidazole in 1× PBS, then washed once with 40 mM imidazole in 1× PBS, and finally eluted with 300 mM imidazole in 1× PBS thrice. Wash and elution fractions were subjected to denaturing sodium dodecyl sulfate-polyacrylamide gel electrophoresis (Fig. S7a). All elution fractions were combined and concentrated using a centrifugal filter unit with a 30 kDa molecular weight cut-off (Millipore) via centrifugation at 4000 × g and 4 °C for 15 min. The concentrated protein mixture was passed through a Superdex 200 XK 16/100 column in 20 mM Tris-HCl pH 8.0 and 150 mM NaCl for size-exclusion chromatography (Fig. S7b, c). Data was collected using ChromLab software v6.1 (Bio-Rad). Fractions corresponding to ~540 kDa were pooled and concentrated using a centrifugal filter unit with a 30 kDa molecular weight cut-off (Millipore) via centrifugation at 4000 × g and 4 °C for 15 min.

## Differential scanning fluorimetry

200 ng/μL of purified S protein and 5× SYPRO orange (Thermo Fisher Scientific) were added into 20 mM Tris-HCl pH 8.0, 150 mM NaCl in optically clear tubes. SYPRO orange fluorescence intensity in relative fluorescence units (RFU) was measured over temperatures ranging from 10 °C to 95 °C using a CFX Connect Real-Time PCR Detection System (Bio-Rad). Melting temperature ($T_m$) was calculated as the temperature at which the first derivative of fluorescence intensity with respect to temperature, $-\frac{d(RFU)}{dT}$, was minimum.

## Reporting summary

Further information on research design is available in the Nature Portfolio Reporting Summary linked to this article.

## Data availability

Structures from the following identifiers from the Protein Data Bank (PDB) were used in this study: 6VXX and 6VYB. The cryoEM map of 2PQ spike can be accessed at the Electron Microscopy Data Bank (EMDB) using accession code EMD-29374. Raw deep sequencing data generated in this study have been submitted to the NIH Sequence Read Archive under accession number: PRJNA826665. Source data are provided with this paper.

## Code availability

Custom codes to analyze deep mutational scanning, thermal stability, and flow cytometry data have been deposited to https://doi.org/10.5281/zenodo.7742830[79].

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

## Acknowledgements

We thank the Roy J. Carver Biotechnology Center at the University of Illinois at Urbana-Champaign for assistance with fluorescence-activated cell sorting and deep sequencing. We thank the cryogenic-electron microscopy core facility at the Case Western Reserve University School of Medicine. This work was supported by National Institutes of Health (NIH) R01 AI167910 (N.C.W.), DP2 AT011966 (N.C.W.), R35 GM142886 (K.A.M.), the Michelson Prizes for Human Immunology and Vaccine Research (N.C.W.), the Searle Scholars Program (N.C.W.), and the Bill and Melinda Gates Foundation INV-004923 (I.A.W.).

## Author contributions

T.J.C.T. and N.C.W. conceived and designed the study. T.J.C.T. established the fusion assay and performed the deep mutational scanning experiments. T.J.C.T. and N.C.W. analyzed the deep mutational scanning data. T.J.C.T., R.L. and W.O.O. expressed and purified recombinant proteins. Z.M. and X.D. performed cryo-EM analysis. K.A.M. provided the landing pad cells and helped establish the fusion assay. M.Y. and I.A.W. provided the CC12.3 antibody; G.S. and R.A. provided the CC40.8 antibody. T.J.C.T. and C.K. performed the microscopy analysis. T.J.C.T. and N.C.W. wrote the paper and all authors reviewed and/or edited the paper.

## Competing interests

N.C.W., K.A.M. and T.J.C.T. have filed a provisional patent application (IL0042US.L) with the University of Illinois covering the deep mutational scanning-based method to identify prefusion-stabilizing mutations for vaccine design and SARS-CoV-2 spike with the K986P/V987P/D994Q mutations described in this article. N.C.W. serves as a consultant for HeliXon. The remaining authors declare no other competing interests.
