## [Peer Review File · Nature Communications]

Reviewers' Comments:

Reviewer #1:

Remarks to the Author:

Tan et. al., describe an interesting high throughput screening method to identify mutations that stabilize the prefusion SARS-CoC-2 spike based on measurement of fusion activity and expression levels for each individual cell which is linked to the identity of the single mutation.

The technology they used which is described in the Nucleic Acid Research paper is very elegant and it may have many diverse applications

Although the screening was done on a relatively small part of a relatively large protein (only 12.5% of ectodomain), such an approach can be applied to the most unstable regions of a protein that are most likely stabilized by point mutations. The 994Q mutation that they found is a very interesting and successful stabilizing mutation.

There are several problems with the current study:

Measurement is based on expression level and fusogenicity but if fusion is impacted and expression is maintained as measured by CC12.3 binding, there is no good control for correct folding of spike. CC12.3 is specific for RBD and is not a good probe for general folding or misfolding of the protein. Correct folding of the non-fusogenic spike could better be probed with another prefusion conformation - dependent antibody. Inhibition of fusogenicity is a good control for fusogenic variants but not for non-fusogenic variants.

Just like the study of by Juraszek et. al., (Nat Comm 2021) mutations at position 943 and 944 increased expression. According to the Supplemental data, many other mutations at positions 943 and 944 did not or hardly increased expression levels (FIG S4) but according to Juraszek et. al., mutations 943G, 944G and especially 944P (10-fold) had a strong impact on the expression. This discrepancy needs to be addressed. The mutations described in this manuscript should be compared with these earlier described mutations on expression level of the soluble and full-length protein. Perhaps this difference is attributed to the difference between the impact on soluble or full-length expression as was seen for the mutation at position 1005 but it certainly needs an explanation and a reference to the previous Nature Communications paper.

If the strong increase in expression level by e.g. 943G, 944G or 944P can't be reproduced with this method, than there is apparently something that is missed in this high throughput screening. If these mutations just don't have an impact on the full-length expression but only on soluble, this phenomenon should be highlighted even stronger in the current discussion since then it is not only Q1005R that has such an effect but the discrepancy might be higher.

It is reassuring to see that the 986P and 987P are picked up. 817P from hexapro is not picked up and also A899P doesn't reach an expression score over 1 so it is not needed to highlight. 942P and 892P are picked up but they should be labeled to both references since these mutations were also discovered by Juraszek et. al. in the Nature Communication paper and should be indicated in Fig 2 a and b (S-Closed and Hexapro).

Although the improvement of 2PQ is clear, the melting temperature is lower than published stabilized versions like in the Hsieh paper. It is published that soluble spike proteins can suffer from cold denaturation and therefore it is probably important to measure the melting temperature directly with fresh material (for both fresh 2P and 2PQ). The very low resolution for CryoEM and limited analytical data make it difficult to appreciate the improvement of 2PQ. The resolution of the structure is so low that modeling had to be used instead in order to try explain the mechanism of the most important stabilizing mutation. So it may be true, but it is hypothetical. Using very fresh protein or inclusion of additional known stabilizing mutations could result in a better CryoEM analysis.

Reviewer #2:

Remarks to the Author:

The strength of this paper is development of a high throughput screening assays for mutations which improve the expression but decrease the fusogen potential for Type 1 viral fusion proteins. The current demonstration of this assay is for Wuhan Hu-1 Spike protein (S); one set of

mammalian cells surface express ACE2 and a portion of a fluorescent protein; another set express S variants and another fluorescent protein. Upon fusion, the cells are larger which can be collected by light scattering measurement as well reconstitution of a fluorescent protein. Combined with an existing expression assay allows one to identify mutations which improve expression but decrease fusogen potential, which is significant for the design of immunogens and diagnostic reagents for new emerging viral pathogens of concern. The authors performed biological replicate measurements of their high throughput assay, showed a variant with higher expression yield and modified stability compared with the common S2P variant, and generated a cryo-EM structure of this new P2Q mutant.

There are a few weaknesses which can mostly be corrected. The authors do not claim that their uncovered P2Q variant (the S2P mutations with D994Q) is more stable than Hexapro or other engineered S proteins, in which case they would need to demonstrate this experimentally. Since the novelty of the paper is the new fusogen screening method, I would appreciate more details of the kinetic limitations of the assay (waiting too long results in unacceptably large cell aggregates that cannot be sorted), the relatively lower reproducibility of the fusion scores (an $r=0.61$, which is quite lower than other mammalian based DMS experiments), perhaps owing to the difficulties of the timing of the experiment to avoid clogging of cell sorters by cell aggregates. The generality of the screen also should be discussed, as for some viruses the cell surface receptor is not known. There are also two experiments in the current version of this manuscript which need to be revisited:

1. Fig 3, Fig S5. replicates are needed to assess statistical significance (n=2 reported, should be n=3).

2. Fig S9 & Figure 4. The BLI fits are not publishable and need to be either redone or fitted to alternative 1:1 binding kinetics, as there are large discrepancy between model fits and grey datapoints. The methods nor figure legends do not state whether IgG or Fabs were measured; I suspect IgG's were used in which case alternative binding kinetics should be fit. This could explain some but not all of the measurement difficulties - I would recommend performing this measurement using Fabs, test different S labeling concentrations (the labeling concentrations reported in the methods seem quite high), and screening solution conditions to minimize baseline drift.

Minor:

Fig S1, Fig 1b - Labeling strategy for hACE2 and S displaying cells , respectively, needs to be clearly identified.

Reviewer #3:

Remarks to the Author:

The manuscript from Tan et al reports a cell-based method to identify prefusion-stabilizing substitutions in the SARS-CoV-2 spike. This a clever method that allows for high-throughput and comprehensive assessment of individual amino acid substitutions. As proof-of-concept, the authors are able to identify known stabilizing proline substitutions that were designed using structure-based approaches. They also identify several novel substitutions that increase expression and stability of SARS-CoV-2 spikes. One of these, D994Q, in combination with the previously identified K986P/V987P substitutions, substantially boosts expression and stability, and a 3.6 Å cryo-EM structure provides insight into its mechanism.

The finding that some of the substitutions identified via the cell-based method do not translate to soluble proteins is intriguing. It suggests a potential short-coming of the system that the authors are encouraged to elaborate on. Substitutions that cause premature triggering of class I viral fusion proteins may result in both an increase in expression and a lack of fusogenicity, which would look promising via the cell-based method yet are the opposite of what one intends to obtain. The authors should provide more balance to the writing, as although this method will be very helpful for the identification of stabilizing substitutions, the substitutions still need to be

experimentally verified on soluble proteins. Overall the manuscript is well written and the results (and method) represent a substantial advance that should be of great interest to many and will complement structure-based methods.

Reviewer #1 (Remarks to the Author):

Tan et. al., describe an interesting high throughput screening method to identify mutations that stabilize the prefusion SARS-CoC-2 spike based on measurement of fusion activity and expression levels for each individual cell which is linked to the identity of the single mutation.

The technology they used which is described in the Nucleic Acid Research paper is very elegant and it may have many diverse applications

Although the screening was done on a relatively small part of a relatively large protein (only 12.5% of ectodomain), such an approach can be applied to the most unstable regions of a protein that are most likely stabilized by point mutations. The 994Q mutation that they found is a very interesting and successful stabilizing mutation.

Response: We thank the reviewer for their positive comments and appreciation of our study.

There are several problems with the current study:

Measurement is based on expression level and fusogenicity but if fusion is impacted and expression is maintained as measured by CC12.3 binding, there is no good control for correct folding of spike. CC12.3 is specific for RBD and is not a good probe for general folding or misfolding of the protein. Correct folding of the non-fusogenic spike could better be probed with another prefusion conformation - dependent antibody. Inhibition of fusogenicity is a good control for fusogenic variants but not for non-fusogenic variants.

Response: We agree with this potential concern and have discussed it in the revised manuscript:

See lines 268-271: "Since RBD is present in the prefusion conformation but not the postfusion conformation⁵⁻⁷ and is the major target of neutralizing antibodies⁶¹, this study used an RBD antibody to probe for surface expression. Nevertheless, we acknowledge that the folding of prefusion S can be more comprehensively probed by antibodies to conformational epitopes in the NTD and S2 subunit."

Just like the study of by Juraszek et. al., (Nat Comm 2021) mutations at position 943 and 944 increased expression. According to the Supplemental data, many other mutations at positions 943 and 944 did not or hardly increased expression levels (FIG S4) but according to Juraszek et. al., mutations 943G, 944G and especially 944P (10-fold) had a strong impact on the expression. This discrepancy needs to be addressed. The mutations described in this manuscript should be compared with these earlier described mutations on expression level of the soluble and full-length protein. Perhaps this difference is attributed to the difference between the impact on soluble or full-length expression as was seen for the mutation at position 1005 but it certainly needs an explanation and a reference to the previous Nature Communications paper.

If the strong increase in expression level by e.g. 943G, 944G or 944P can't be reproduced with this method, than there is apparently something that is missed in this high throughput screening. If these mutations just don't have an impact on the full-length expression but only on soluble, this phenomenon should be highlighted even stronger in the current discussion since then it is not only Q1005R that has such an effect but the discrepancy might be higher.

Response: We do think this discrepancy between membrane-bound and soluble proteins deserves more attention and have emphasized it in the revised discussion:

See lines 243-250: “One interesting finding in this study is that the expression of membrane-bound (i.e. full-length) S protein does not necessarily correlate with the expression of soluble S ectodomain, as exemplified by Q1005R. In addition, our results show that I943G, D944G and D944P mutations, which have been shown to increase the expression of soluble S ectodomain²⁵, do not increase the expression of membrane-bound S protein. These observations indicate that the ectodomain of the S protein has some long-range interactions with its native transmembrane domain. As a result, caution is needed when extrapolating the results obtained from full-length S protein to soluble S ectodomain, or vice versa.”

It is reassuring to see that the 986P and 987P are picked up. 817P from hexapro is not picked up and also A899P doesn't reach an expression score over 1 so it is not needed to highlight. 942P and 892P are picked up but they should be labeled to both references since these mutations were also discovered by Juraszek et. al. in the Nature Communication paper and should be indicated in Fig 2 a and b (S-Closed and Hexapro).

Response: We have updated the labeling to incorporate S-Closed. We have added the corresponding citation in the legend of Figure 2:

“Mutations used in S-Closed²⁵ and/or HexaPro²⁴ are in yellow.”

Although the improvement of 2PQ is clear, the melting temperature is lower than published stabilized versions like in the Hsieh paper. It is published that soluble spike proteins can suffer from cold denaturation and therefore it is probably important to measure the melting temperature directly with fresh material (for both fresh 2P and 2PQ).

Response: Thank you for point this out. In the revised manuscript, we have acknowledged that D994Q was not as stabilizing as a combination of F817P, A892P, A899P, A942P that were used in HexaPro (Hsieh et al. 2020).

See lines 207-210: “The stabilizing effect of D994Q, however, was not as pronounced as a combination of F817P, A892P, A899P, A942P that were used in HexaPro, which not only showed two peaks in the differential scanning fluorimetry analysis, but also shifted the first apparent melting temperature by 5 °C²⁴.

Since the melting temperature measurements were done very shortly after the proteins were purified and EM analysis also showed that the proteins folded properly, we do not think cold denaturation is playing a significant role here.

The very low resolution for CryoEM and limited analytical data make it difficult to appreciate the improvement of 2PQ. The resolution of the structure is so low that modeling had to be used instead in order to try explain the mechanism of the most important stabilizing mutation. So it may be true, but it is hypothetical. Using very fresh protein or inclusion of additional known stabilizing mutations could result in a better CryoEM analysis.

Response: We agree with the reviewer that the cryoEM structure prevents us to draw conclusion at the atomic level. However, it still enabled us to confirm that the 2PQ mutant is in the prefusion-stabilized conformation. In addition, the electron density of the protein backbone is clear and enabled us to draw conclusions about the backbone positioning. Specifically, we compared our cryoEM density map of 2PQ to that of an existing structure with 2P (PDB: 6VXX) and investigated the helix that contained D/Q944 (Fig. S8b). Helices are closer in 2PQ compared to those in 2P, which could account for the increased stability of the prefusion configuration of S with the 2PQ mutations. We have added the following in the cryoEM results subsection:

See lines 197-200: "While the structure could not be resolved at the atomic resolution, the result allowed us to confirm that the 2PQ mutant is in the prefusion-stabilized conformation. In addition, the electron density of the protein backbone clearly showed that the helix containing residue 944 is shifted towards the helix containing residue Q755 (Fig. S8b)."

We appreciate the reviewer's suggestions of strategies to improve the cryoEM analysis, although we failed to achieve atomic resolution after multiple attempts. We are continuing to explore alternative strategies for cryoEM analysis on spike mutants and currently aiming to characterize other previously unknown prefusion-stabilizing mutations. We hope to present the results in a separate study.

Reviewer #2 (Remarks to the Author):

The strength of this paper is development of a high throughput screening assays for mutations which improve the expression but decrease the fusogen potential for Type 1 viral fusion proteins. The current demonstration of this assay is for Wuhan Hu-1 Spike protein (S); one set of mammalian cells surface express ACE2 and a portion of a fluorescent protein; another set express S variants and another fluorescent protein. Upon fusion, the cells are larger which can be collected by light scattering measurement as well reconstitution of a fluorescent protein. Combined with an existing expression assay allows one to identify mutations which improve expression but decrease fusogen potential, which is significant for the design of immunogens and diagnostic reagents for new emerging viral pathogens of concern. The authors performed biological replicate measurements of their high throughput assay, showed a variant with higher expression yield and modified stability compared with the common S2P variant, and generated a cryo-EM structure of this new P2Q mutant.

Response: We thank the reviewer for their positive comments and appreciation of our study.

There are a few weaknesses which can mostly be corrected. The authors do not claim that their uncovered P2Q variant (the S2P mutations with D994Q) is more stable than Hexapro or other engineered S proteins, in which case they would need to demonstrate this experimentally.

Response: The stability of HexaPro was measured in a previous study (Hsieh et al. 2020) by differential scanning fluorimetry, which is the same technique that was used in our present study. In the revised manuscript, the stability effect of the quadruple mutant F817P/A892P/A899P/A942P (HexaPro, result from Hsieh et al. 2020) was compared with that of D994Q (P2Q):

See lines 207-210: “The stabilizing effect of D994Q, however, was not as pronounced as a combination of F817P, A892P, A899P, A942P that were used in HexaPro, which not only showed two peaks in the differential scanning fluorimetry analysis, but also shifted the first apparent melting temperature by 5 °C²⁴.

Since the novelty of the paper is the new fusogen screening method, I would appreciate more details of the kinetic limitations of the assay (waiting too long results in unacceptably large cell aggregates that cannot be sorted), the relatively lower reproducibility of the fusion scores (an $r=0.61$, which is quite lower than other mammalian based DMS experiments), perhaps owing to the difficulties of the timing of the experiment to avoid clogging of cell sorters by cell aggregates.

Response: The reviewer is correct about the need to avoid clogging of cell sorters by cell aggregates. In the revised manuscript, we have acknowledged this limitation in the discussion and proposed a potential solution that is currently being explored in our lab:

See lines 272-276: “Furthermore, due to the technical difficulties in sorting large syncytia, co-culturing of S-expressing cells and hACE2-expressing cells could only be performed for relatively short durations before sorting, leading to fewer syncytia and potentially lower reproducibility of results across replicates. Alternative strategies including microfluidics-based fusion experiments can be explored to obviate the kinetic limitations of the current fusion assay.”

The generality of the screen also should be discussed, as for some viruses the cell surface receptor is not known.

Response: We have added this salient point in our revised discussion:

See lines 237-239: “While we only provide a proof-of-concept using the SARS-CoV-2 S protein, our approach can be adopted to fusion proteins of other viruses with known cell surface receptors.”

There are also two experiments in the current version of this manuscript which need to be revisited:

1. Fig 3, Fig S5. replicates are needed to assess statistical significance (n=2 reported, should be n=3).

Response: We have updated **Fig. 3** and **Fig. S5** by performing one more independent biological replicate and a two-sided Welch's *t*-test.

2. Fig S9 & Figure 4. The BLI fits are not publishable and need to be either redone or fitted to alternative 1:1 binding kinetics, as there are large discrepancy between model fits and grey datapoints. The methods nor figure legends do not state whether IgG or Fabs were measured; I suspect IgG's were used in which case alternative binding kinetics should be fit. This could explain some but not all of the measurement difficulties - I would recommend performing this measurement using Fabs, test different S labeling concentrations (the labeling concentrations reported in the methods seem quite high), and screening solution conditions to minimize baseline drift.

Response: Thank you for the technical suggestions. However, after multiple attempts, measurement difficulties with BLI remained unresolved. As a result, we performed titration of the antibodies via flow cytometry for binding affinity comparison (**Fig. 4e, Fig. S9**). We have removed text regarding BLI and have replaced them with the following:

See lines 212-216 in the Results section: "We compared the binding of 2P and 2PQ to various S antibodies, including CC12.3 (RBD)⁴⁴, S2M28 (NTD)⁴⁸, CC40.8 (S2 stem helix)⁴⁵, and COVA1-07 (S2 HR1)⁴⁹, using flow cytometry. 2P and 2PQ showed similar binding affinity to CC12.3 and S2M28 (**Fig. 4e, Fig. S9a,b**). However, when assayed for binding with COVA1-07 or with CC40.8, 2PQ had weaker binding than 2P (**Fig. 4e, Fig. S9c,d**)."

See lines 218-220 in the Results section: "Similarly, the binding of CC40.8 to S requires partial disruption of the prefusion S trimer and is shown to be weakened by prefusion-stabilizing mutations⁴⁵."

See lines 403-409 in the Methods section: "For titration of S-2P or S-2PQ, HEK293T landing pad cells stably expressing membrane-bound S-2P or S-2PQ were incubated with 0, 0.1, 0.3, 1.0, 3.0 or 10.0 µg/mL of S2M28, CC12.3, COVA1-07, or CC40.8 antibody for 1 h in ice-cold FACS buffer. Cells were washed and then incubated with 1 µg/mL PE-conjugated anti-human IgG Fc for 1 h at 4 °C. Cells were washed, resuspended in ice-cold FACS buffer, and analyzed for levels of PE using an Accuri C6 flow cytometer. Gating strategy is shown in **Fig. S11a**. MFI values were subtracted from those of negative control (0 µg/mL of antibody) and plotted against antibody concentration (**Fig. 4e, Fig. S10**)."

Minor:

Fig S1, Fig 1b - Labeling strategy for hACE2 and S displaying cells, respectively, needs to be clearly identified.

Response: We have added text in the figure legend regarding the primary antibody used for labeling these cells:

See figure legend of Fig. 1b: “Primary antibody used was CC12.3, an RBD antibody⁴⁴.”

See figure legend of Fig. S1: “Primary antibody used was S RBD with an Fc tag.”

Reviewer #3 (Remarks to the Author):

The manuscript from Tan et al reports a cell-based method to identify prefusion-stabilizing substitutions in the SARS-CoV-2 spike. This a clever method that allows for high-throughput and comprehensive assessment of individual amino acid substitutions. As proof-of-concept, the authors are able to identify known stabilizing proline substitutions that were designed using structure-based approaches. They also identify several novel substitutions that increase expression and stability of SARS-CoV-2 spikes. One of these, D994Q, in combination with the previously identified K986P/V987P substitutions, substantially boosts expression and stability, and a 3.6 Å cryo-EM structure provides insight into its mechanism.

Response: We thank the reviewer for their positive comments and appreciation of our study.

The finding that some of the substitutions identified via the cell-based method do not translate to soluble proteins is intriguing. It suggests a potential short-coming of the system that the authors are encouraged to elaborate on. Substitutions that cause premature triggering of class I viral fusion proteins may result in both an increase in expression and a lack of fusogenicity, which would look promising via the cell-based method yet are the opposite of what one intends to obtain. The authors should provide more balance to the writing, as although this method will be very helpful for the identification of stabilizing substitutions, the substitutions still need to be experimentally verified on soluble proteins. Overall the manuscript is well written and the results (and method) represent a substantial advance that should be of great interest to many and will complement structure-based methods.

Response: We have added text regarding the differences in membrane-bound and soluble proteins in our revised discussion:

See lines 243-250: “One interesting finding in this study is that the expression of membrane-bound (i.e. full-length) S protein does not necessarily correlate with the expression of soluble S ectodomain, as exemplified by Q1005R. In addition, our results show that I943G, D944G and D944P mutations, which have been shown to increase the expression of soluble S ectodomain²⁵, do not increase the expression of membrane-bound S protein. These observations indicate that the ectodomain of the S protein has some long-range interactions with its native transmembrane domain. As a result, caution is needed when extrapolating the results obtained from full-length S protein to soluble S ectodomain, or vice versa.”

Nevertheless, we do not think mutations that prematurely convert to postfusion conformation would show as high expression in our screen, because we used an RBD antibody to probe for expression and RBD is only present in the prefusion conformation, but not postfusion conformation. This is clarified in the manuscript:

See lines 268-270: "Since RBD is present in the prefusion conformation but not the postfusion conformation⁵⁻⁷ and is the major target of neutralizing antibodies⁶¹, this study used an RBD antibody to probe for surface expression."

Reviewers' Comments:

Reviewer #1:

Remarks to the Author:

Revisions are good but something went wrong with Figure 2 a and b it seems. the doc file and pdf missed the grey and colored dots.

march 6th:

figure looks ok now

Reviewer #2:

Remarks to the Author:

The authors have addressed and satisfied all concerns with the original paper.

Reviewer #1 (Remarks to the Author):

Revisions are good but something went wrong with Figure 2 a and b it seems. the doc file and pdf missed the grey and colored dots.

march 6th:
figure looks ok now

Response: We thank the reviewer for the positive comments.

Reviewer #2 (Remarks to the Author):

The authors have addressed and satisfied all concerns with the original paper.

Response: We thank the reviewer for the positive comments.